# Human Genome Polymorphisms and Computational Intelligence Approach Revealed a Complex Genomic Signature for COVID-19 Severity in Brazilian Patients

**DOI:** 10.3390/v15030645

**Published:** 2023-02-28

**Authors:** André Filipe Pastor, Cássia Docena, Antônio Mauro Rezende, Flávio Rosendo da Silva Oliveira, Marília de Albuquerque Sena, Clarice Neuenschwander Lins de Morais, Cristiane Campello Bresani-Salvi, Luydson Richardson Silva Vasconcelos, Kennya Danielle Campelo Valença, Carolline de Araújo Mariz, Carlos Brito, Cláudio Duarte Fonseca, Cynthia Braga, Christian Robson de Souza Reis, Ernesto Torres de Azevedo Marques, Bartolomeu Acioli-Santos

**Affiliations:** 1Sertão Pernambucano Federal Institute of Education, Science and Technology, Petrolina 56316-686, PE, Brazil; 2Department of Microbiology, Icahn School of Medicine at Mount Sinai, New York, NY 10029-6574, USA; 3Core Facility, Oswaldo Cruz Foundation, Recife 50740-465, PE, Brazil; 4Department of Microbiology, Aggeu Magalhães Institute, Oswaldo Cruz Foundation, Recife 50740-465, PE, Brazil; 5Federal Institute of Education, Science and Technology of Pernambuco, Recife 50740-545, PE, Brazil; 6Department of Virology, Aggeu Magalhães Institute, Oswaldo Cruz Foundation, Recife 50740-465, PE, Brazil; 7Department of Parasitology, Aggeu Magalhães Institute, Oswaldo Cruz Foundation, Recife 50740-465, PE, Brazil; 8Department of Clinical Medicine, Pernambuco Federal University, Recife 50740-600, PE, Brazil; 9Servidores do Estado Hospital (HSE), Recife 52020-020, PE, Brazil; 10Department of Infectious Diseases and Microbiology, Graduate School of Public Health, University of Pittsburgh, Pittsburgh, PA 15261, USA

**Keywords:** COVID-19 genetics, SARS-CoV-2 infection, complex genomic classifier, machine learning

## Abstract

We present a genome polymorphisms/machine learning approach for severe COVID-19 prognosis. Ninety-six Brazilian severe COVID-19 patients and controls were genotyped for 296 innate immunity loci. Our model used a feature selection algorithm, namely recursive feature elimination coupled with a support vector machine, to find the optimal loci classification subset, followed by a support vector machine with the linear kernel (SVM-LK) to classify patients into the severe COVID-19 group. The best features that were selected by the SVM-RFE method included 12 SNPs in 12 genes: *PD-L1*, *PD-L2*, *IL10RA*, *JAK2*, *STAT1*, *IFIT1*, *IFIH1*, *DC-SIGNR*, *IFNB1*, *IRAK4*, *IRF1*, and *IL10*. During the COVID-19 prognosis step by SVM-LK, the metrics were: 85% accuracy, 80% sensitivity, and 90% specificity. In comparison, univariate analysis under the 12 selected SNPs showed some highlights for individual variant alleles that represented risk (*PD-L1* and *IFIT1*) or protection (*JAK2* and *IFIH1*). Variant genotypes carrying risk effects were represented by *PD-L2* and *IFIT1* genes. The proposed complex classification method can be used to identify individuals who are at a high risk of developing severe COVID-19 outcomes even in uninfected conditions, which is a disruptive concept in COVID-19 prognosis. Our results suggest that the genetic context is an important factor in the development of severe COVID-19.

## 1. Introduction

In recent years, humanity has faced the COVID-19 pandemic, an infectious disease caused by the SARS-CoV-2 virus. The first COVID-19 patients were documented in December of 2019 in China, and now the virus has spread around the world, causing a pandemic with over 668,000,000 cases and over 6.7 million deaths reported by January 2023 [1]. The SARS-CoV-2 infection ranges from asymptomatic to life-threatening, wherein the most common symptoms are fever, cough, diarrhea, shortness of breath, pneumonia, and acute respiratory distress [2,3]. The severe disease especially affects patients with comorbidities such as diabetes and cardiovascular illnesses, which are commonplace in the elderly population [2,4]. In addition to comorbidities and age, studies have described important factors that are related to the severe progression of COVID-19, such as virus genetics [5], sex [6,7], and host genetics [4,8].

Previous works have focused on predicting the importance of variations in host genes in the interplay between the SARS-CoV-2 infection and COVID-19 progression. A comparison of the *ACE2* gene using whole exome sequencing (WES) data identified higher variability in the controls compared with COVID-19 patients in Italy [9]. *ACE2* or *TMPRSS2* DNA polymorphisms offered potential explanations for differential genetic susceptibility to COVID-19 across eight populations [10]. Consistent with this, soluble ACE2, K26R. and T92I was more effective in blocking the entry of the S protein-pseudo-typed virus, suggesting that *ACE2* variants can modulate their susceptibility to SARS-CoV-2 [11].

After the first initiative, genome-wide association studies were performed using data from different populations. A total of 1980 patients with COVID-19 and severe disease from Italy and Spain were analyzed, and it was ascertained that the 3p21.31 gene cluster is a genetic susceptibility locus in COVID-19 patients with respiratory failure. Additionally, a potential involvement of the *SLC6A20* gene and the ABO blood group system was confirmed [12]. From China, 339 COVID-19 patients were sequenced with a gene locus that was associated with severity located in *TMEM189–UBE2V1* and which was involved in the IL-1 signaling pathway. Moreover, the *HLA* A*11:01, B*51:01, and C*14:02 alleles significantly predisposed the worst outcome for the infected individuals [13]. From 208 intensive care units in the UK, 2244 critically ill patients with COVID-19 were found to have variations in genes relating to antiviral response, such as *IFNAR2* and *OAS1*, *2*, and *3* [14]. A Brazilian study investigated 27 candidate genes and HLA alleles in 954 exomes from two Brazilian databases. The results showed the presence of six nonsynonymous variants that influenced the infection rate or the clinical prognosis of COVID-19, in addition to HLA alleles at the DQB1 and DRB1 loci which had previously been associated with the human antiviral response against SARS-CoV-2 [15]. Three genome-wide association meta-analyses consisting of 49,562 COVID-19 patients from 46 studies across 19 countries were performed. This study reported nine genome-wide significant loci in relation to lung or autoimmune and inflammatory diseases associated with severe manifestations of COVID-19 [5]. The hemizygous missense variant in *XIAP* and hemizygous missense variant in *CYBB* (p. Arg229Thr), which encodes the p91phox subunit of the NADPH oxidase, were related to the multisystem inflammatory syndrome in children (MIS-C) with COVID-19 [16].

Recent works have added artificial intelligence (AI) analysis, such as deep learning (DL) and machine learning (ML), to improve the understanding of COVID-19 diagnosis and progression [17,18]. Deep learning application examples include (1) a study that used DL to create a chest-X-ray-image standardization for COVID-19 pneumonia diagnosis and severity compared to healthy and viral-infected groups [19]. For this work, U-Nets, fully convolutional networks, and DeepLabv3 were used, while the latter showed the best performance. Additionally, (2) a study that established a system using computational tomography (CT) scans and clinical data to predict COVID-19 malignant progression was successful in both local and multicenter evaluations [20]. Here, a custom architecture is proposed, combining multilayer perceptron, Resnet, and LSTMs for classification. Some other studies utilized machine learning to investigate severe biochemical, biological, and genetic COVID-19 aspects [8,21,22]. One of them was based on biological features such as lipids, metabolites, proteins, and mRNA using an extreme gradient boosting (XGBoost) model to classify COVID-19 severity [22]. Another approach used five SNPs, including sex and age, to classify COVID-19 patients by applying a multilayer perceptron with two hidden layers [8]. Meanwhile, a third study used machine learning technology to examine WES data from five different cohorts (Germany, Italy, Quebec, Sweden, and the UK). It revealed that immune genes, such as *TLR7*, *TLR3*, *TICAM1*, *TLR8*, *IRAK*, and *RnaseL*, were associated with COVID-19 severity [21]. For this paper, a LASSO logistic regression model was used to identify the most informative variants for severe or mild cases. These data suggest a broad complexity in the genetic markers of patients with COVID-19 disease.

COVID-19 disease is a multi-causal/multi-loci complex disease. We believe that these kinds of genomic rearrangements can only be observed under mass genotyping and machine learning interpretation. Therefore, we present a human genome polymorphism data/machine learning approach that can be used to produce a potential genome prognosis tool for several COVID-19 phenotypes. Our classifier was developed using a panel of 191 SNPs from 96 Brazilian patients with mild and severe COVID-19. The purpose was to assess a set of SNPs that were capable of identifying patients at risk of severe disease. We found an optimal 12 SNPs/genes panel that was able to predict severe COVID-19 with high accuracy, sensitivity, and specificity, using SNPs/genes from viral recognition and antiviral responses.

## 2. Material and Methods

Our method using human genome polymorphisms/machine learning techniques to determine severe COVID-19 prognosis could be separated into three steps: 1. data acquisition, 2. data preprocessing, and 3. data analysis and prognosis.

### 2.1. Data Acquisition

#### 2.1.1. Patient Group

Ninety-six COVID-19-positive patients were enrolled in this study: forty-eight of them with mild and the other forty-eight with severe COVID-19 symptoms. Two hospitals were used for patient recruitment in the city of Recife, Brazil: Hospital dos Servidores do Estado de Pernambuco and Real Hospital Português. The patients were invited to participate in this research, and after a thorough explanation of the project, the individuals who consented to participate were enrolled in our study (ethics committee approbation n° CAAE: 36403820.2.0000.5190 and CAAE: 38435120.5.0000.5190). Patients were categorized as having mild COVID-19 when they had a positive qRT-PCR without the severe symptoms described below. Patients with severe COVID-19 were those who had a positive qRT-PCR with at least one of the following phenotypes: hospital care and mechanical support ventilation (non-invasive ventilation, high-flow oxygen, intubation, and mechanical ventilation, ECMO—extracorporeal membrane oxygenation, RRT—renal replacement therapy, etc.), oxygen saturation under 96%, or death. Vaccination against SARS-CoV-2 was considered an exclusion criterion.

Whole blood collected from the patients was processed and submitted to cryopreservation under −80 °C.

#### 2.1.2. Genomic DNA Extraction

The genomic DNA extraction was performed using the whole blood of patients through the illustra blood genomicPrep Mini Spin Kit (GE Healthcare, Chicago, IL, USA) and the PureLink^®^ Genomic DNA Kit (Invitrogen, Waltham, MA, USA) following the manufacturer’s protocols. NanoDrop 2000/2000c Spectrophotometer (Thermo Scientific, Waltham, MA, USA) and Qubit^®^ 3.0 Fluorometer (Life Technologies, Carlsbad, CA, USA) were utilized to measure the DNA concentration and purity based on 260 nm/280 nm absorbance ratio. The best sample from each patient was chosen to proceed to the next steps.

#### 2.1.3. DNA Library Preparation and Genotyping

An AmpliSeq custom DNA panel for Illumina (number 20019102) was accomplished online (Illumina DesignStudio tool; https://designstudio.illumina.com/ (accessed on 1 April 2021)) (Appendix A). Our panel of SNPs was personally selected by researchers from the Oswaldo Cruz Foundation, and collaborators were custom-developed for the genetic study of human viral diseases. We chose polymorphisms (SNPs) that had already been related in the literature, at some time and population, with the development of some viral diseases. These SNPs are usually from genes that are involved with the innate immune system or that act directly on antiviral responses. This panel has been used by our group in studies with other viral diseases, such as dengue [23].

A total of 15 ng of DNA per sample was used for target enrichment by a multiplex PCR reaction, which was designed for 283 amplicons targeting 296 SNPs in one pool. After 17 PCR cycles, the FuPa reagent was used to digest primer dimers and partially digest PCR amplicons. The unique index combination for dual-index-tagged libraries was generated for each sample using the AmpliSeq library preparation kit with 96 CD-indexes according to the manufacturer’s instructions (see AmpliSeq for Illumina on-demand, custom, and community panels’ reference guide; document #1000000036408, v09). The barcoded libraries were quantified with the Qubit^®^ 3.0 Fluorometer (Life Technologies) and normalized for DNA concentration to 12 multiplexed library pools. To determine the Molar concentration of the 12 multiplexed pools, a qPCR standard library quantification was performed using the ProNex^®^ NGS Library Quant Kit (Promega catalog number NG1201). Agarose gel was used to determine the size of PCR products. Each multiplexed pool and a phiX spike-in were combined to a final loading concentration of 19 pM, which was sequenced on Illumina MiSeq using the MiSeq Reagent v3 for 600 cycles in a single 2 × 150 base pair run.

### 2.2. Data Preprocessing

The 296 polymorphisms for the 96 patient samples were sequenced. To analyze the data, first, all sequencing results were assessed with the FASTQC tool version 0.11.8 (https://www.bioinformatics.babraham.ac.uk/projects/fastqc/, accessed on 1 April 2021). The quality results were compiled using the MultiQC tool version 1.7 [24]. Afterward, the results from the quality control step were used to guide the trimming and filtering step by applying the Trimmomatic tool version 0.38 [25].

Following the sequencing quality evaluation step, the sequencing data were mapped against the human genome (GRCh38) using the bwa tool version 0.7.12 along with the BWA-MEM algorithm [26,27]. After the mapping stage, the variant discovering analysis was performed using the Genome Analysis ToolKit (GATK) version 4.2.2.0 [27]. For this, the best practice workflow for germline variant discovery was applied.

At the end of the previous step, we had 96 VCF files with high-quality variants, which were combined using the GATK command called CombineGVCFs, and then, to perform joint genotyping on the samples pre-called with HaplotypeCaller, we used the GenotypeGVCFs command, which was also from GATK. Finally, the variants were annotated using BCFTOOLS version 1.14 (http://samtools.github.io/bcftools/ (accessed on 10 October 2021)) together with the annotate command and the file with the known human variants cited earlier. The last step was to produce a table of genotypes for the loci of interest, which was performed using the VCF tools version 0.1.13 [28] together with the parameter and argument-extract FORMAT-info and GT, respectively.

### 2.3. Data Analysis and Prognosis

In this step, each locus data were labeled and encoded into integer values considering a categorical scheme of genotypes comprising reference homozygous, heterozygous, or variant homozygous. A total of 105 SNPs were removed from the genome data for having more than 10% missing data. This threshold was considered adequate for balancing the amount of required data imputation and the remaining SNPs for analysis. The remaining 191 SNPs were subject to the missing data imputation most frequently [29]. This approach was selected due to its simplicity and the fact that, biologically, it might make the separation of classes harder while preserving a reasonable number of SNPs to be analyzed. After the preprocessing phase, the dataset kept 191 SNPs and the same initial 96 samples.

### 2.4. Feature Selection

The feature selection phase was necessary because, after the preprocessing phase, the number of samples was still smaller than the number of SNPs. This kind of scenario, in most cases, does not allow for the proper training of machine learning algorithms, which is called the curse of dimensionality [30].

This phase was conducted according to the bootstrap sampling method:

1-For each of the 1000 rounds, the full dataset was resampled with repositioning to generate a training dataset with 96 samples. The test dataset was composed of out-of-bag samples and those not included in the training dataset.

2-For each round, the training and test datasets were submitted to the SVM-RFE algorithm for feature selection. The selected SNPs for each round were registered.

3-The SNPs selected in more than 500 rounds were considered good discriminators of mild vs. severe COVID-19 cases and were used in the patient prognosis phase.

Recursive feature elimination (RFE) is a consolidated technique that is used in feature selection tasks. The main idea behind SVM-RFE [31] is to train the SVM, evaluate feature importance according to this classifier, and recursively remove the least important feature. We decided to use this technique because it is well consolidated (launched in 2002), created to be used in similar cases, and because of previous works of the group [30]. All experiment scripts were implemented using the Python language, and the employed SVM-RFE algorithm was part of the freely downloadable scikit-learn library provided by Pedregosa et al. [32].

### 2.5. Patient Prognosis

To quantitatively evaluate the discriminatory potential of the selected SNPs in the previous phase, five machine-learning techniques with different learning strategies were evaluated. The selected techniques were logistic regression [33], K nearest neighbors [34], decision trees [35], and support vector machines [36], with linear and radial basis functions as kernels, all available in the scikit-learn library. The hold-out validation strategy was used so that each model was trained with 76 samples and tested with 20 distinct samples, which were not seen during the training phase. SHAP (SHapley Additive exPlanations) [37] graphics were developed to quantify the contribution that each feature brought to the prediction made by an ML model. In the current study, that meant the impact of each SNP on the tendency of protection or on the risk of severe COVID-19.

Data analysis for conventional association tests allelic/genomic frequencies were estimated by the software pLINK v. 1.07 on previously selected SNPs in the complex analysis. The existence of associations between groups was evaluated by Chi-square tests, or Fisher’s exact, when appropriate. The differences were considered significant for *p* < 0.05. The magnitude of these associations was estimated as the odds ratio (OR) using 95% confidence intervals.

## 3. Results

### 3.1. COVID-19 Patient Group

The patient group comprised Brazilian individuals who tested positive for SARS-CoV-2 and were admitted to two hospitals in the city of Recife, State of Pernambuco, Brazil, before vaccination efforts in Brazil, between 22 August 2020 and 25 August 2021. In this period, the main local SARS-CoV-2 circulating strains were the P1 (gamma), the AY.99.2 (Delta), the BA.1 (Omicron), and the BA.2 (Omicron). The age of the patients ranged from 11 to 90 years, with 48 non-severe COVID-19 patients and 48 patients with severe clinical phenotypes (Table 1).

### 3.2. Genomic Aspects

A total of 283 amplicons were sequenced for 96 patient samples. The sequencing data had a mean GC content equal to 41.14%. At the end of the sequencing run, 19.1 million paired-end reads were produced, 72.9% of the data had a base quality that was equal to or higher than phred 30, and the mean quality was equal to 31.3. The number of reads per sample ranged from 13,145 to 742,829. The sequencing depth for each locus ranged from 2 to 5075 times.

### 3.3. COVID-19 Genomic Classifier

Our genome polymorphisms/machine learning COVID-19 prognosis classifier, whose features are selected by the SVM-RFE method, including 12 SNPs in 12 innate immune genes (Figure 1). The SNPs and genes are described below: rs1990760 (*IFIH1*), rs2161525 (*DC-SIGNR*), rs4251513 (*IRAK4*), rs3024498 (*IL10*), rs2508450 (*IL10RA*), rs16923189 (*PD-L2*), rs17804441 (*PD-L1*), rs1051922 (*IFNB1*), rs12340866 (*JAK2*), rs3771300 (*STAT1*), rs303215 (*IFIT1*) and rs17622656 (*IRF1*).

In the conventional analyses, individual OR was analyzed for each variant allele/genotype that was present in the genome complex classifier (Figure 2). For the variant allele effect, significant analyses were observed for the risk of severe COVID-19 development in *PD-L1*/rs17804441 SNP (allele variant C, OR = 1.92, CI 1.12–3.45, *p* = 0.045) and *IFIT1*/rs303215 (variant allele C, OR = 3.34, CI 1.55–7.20, *p* = 0.009) loci. On the other hand, the variant allele protection effect was identified as *JAK2*/rs12340866 (variant allele A, OR = 0.48, CI 0.26–0.89, *p* = 0.048) and *IFIH1*/rs1990760 (variant allele T, OR = 0.55, CI 0.34–0.89, *p* = 0.04). For the variant genotype analyses, significant risk effects were identified for the *PD-L2*/rs17804441 genotype CC/TT (OR 2.6, CI 1.08–6.15, *p* = 0.03) for the dominant model and *IFIT1*/rs303215 for the genotype model *p* = 0.04. It was observed that some variant genotypes (rs17622656, rs2508450, and rs303215) were out of Hardy–Weinberg equilibrium in all COVID-19 groups (*n* = 96).

For the complex approach, among the five tested machine-learning techniques (Appendix A), the best classifier was produced by SVM with the linear kernel (SVM-LK). The best performing SVM used the hyperparameters C = 1 and kernel = ‘linear’. All other hyperparameters had defaulted scikit-learn values. The performance indicators of this best complex classifier are shown in Figure 2, with a high sensibility (80%), specificity (90%), and accuracy (85%) for the complex (combined) genome classifier. Since the best-performing complex classifier was produced by an SVM with a linear kernel, it is not possible to directly understand how each input led to each prediction nor the impact that each feature had on the general classification. Therefore, to properly understand the inner functioning of the complex classifier, a state-of-the-art ML explainability technique was used: the SHAP technique [37]. Briefly, SHAP quantifies the contribution that each feature brings to the prediction made by an ML model.

The graph contained in Figure 1 is an adapted version of a violin plot produced by the SHAP API compatible with Scikit-learn. In this figure, it is possible to observe that (i) each one of the 12 selected SNPs was sorted with the most important ones for model classification at the top; (ii) reference homozygotes are represented as blue dots, heterozygotes are represented as purple dots, and alternative homozygous are represented as pink dots; (iii) each dot represents the genotype of a given patient for the row contained within each horizontal line; and (iv) at the bottom, it is possible to see the impact of each SNP on the prediction of mild or severe cases of the complex classifier. It is possible to observe the high relevance of *IFNB1*, *DC-SIGNR*, *JAK2*, *PD-L1*, *IFIH1*, *STAT1*, and *IL10RA* polymorphisms.

It is also interesting to note the “individual” effect of each polymorphic *locus* on the classifier: for some SNPs, the reference homozygous genotype contributed as a risk element (*IFNB1*, rs1051922) and, for others, the variant homozygous genotypes were the risk factors (*STAT1*, rs3771300). Curiously, some heterozygous genotypes worked on both sides of the classification, depending on the genomic context of each patient (i.e., purple heterozygous genotype in *IFIH1*, *STAT1*, and *IL10RA*): a probable codominance effect.

## 4. Discussion

### 4.1. COVID-19 Genomic Prognosis Classifier

The SARS-CoV-2 infection promotes broad asymptomatic to life-threatening symptoms that can culminate in acute respiratory distress syndrome and death as COVID-19 progresses [2,3]. The severe COVID-19 disease especially affects patients with comorbidities such as diabetes and cardiovascular illnesses, which are both commonplace in older populations [2]. In addition to comorbidities and age, studies have described important factors that relate to the progression of severe COVID-19, such as virus genetics [5], sex [6,7], blood type [38], and host genetics [8]. However, the prognosis of COVID-19 has still been a challenge due to the complex nature of the disease. Early prediction of COVID-19 severity is fundamental for better management, patient care, and therapy. In clinical routines, the optimal medical practice would be to identify, as soon as possible, patients that would develop severe forms of the disease and to provide specific and individual treatment to avoid disease progression.

The most proposed COVID-19 classification methods use a wide variety of clinical and laboratory data (disease stage-dependent, tissue-dependent, etc.) and/or subjective data-dependent, sometimes based on clinician experience [22,39]. However, some genetic studies have proposed classification methods for early COVID-19 prediction, avoiding the imprecision of clinical interpretation. Some of these works were based on conventional genetic analysis (i.e., considering individual genome markers effect) [14,40,41], but more recent efforts have attempted new alternatives based on massive genome data and computational intelligence tools. These new ways consider some complex genome signatures that are impossible to be observed under traditional perspectives, which is an emergent field in genetics (see [8]).

In our present work, however, we solely compare human innate immune genome marker (SNPs) data by (i) statistical genetic analysis with (ii) a complex machine learning genome polymorphisms classifier. In the univariate analysis, we observed that variant alleles in *PD-L1*(“C”, rs17804441, OR = 1.92) and in *IFIT1* (“C”, rs303215, OR = 3.34) genes had significant risk effects on the development of severe COVID-19, whereas polymorphism in *JAK2* (“A”, rs12340866, OR = 0.48) and in *IFIH1* (“T”, rs1990760, OR = 0.53) were related to protection. For genotype associations, we observed that *PD-L2* CC/TT (rs17804441, OR = 2.6) in the dominant model and *IFIT1* (rs303215) under the genotype model were highlighted as key elements for severe phenotype development. Some of these SNPs were observed as important elements in SARS-CoV-2 infection or other respiratory viral infections, as discussed below. It is important to note that some non-significant alleles/genotypes in the univariate analysis were considered significant members in the complex classifier (Figure 1), showing the importance of taking a complex view to better understand complex diseases.

The complex genome COVID-19 prognosis classifier, on the other hand, is based on the human genome markers’ complex arrangements data which can be observed by computational intelligence in accordance with the univariate analysis. The SNPs and genes are described below: rs1990760 (*IFIH1*), rs2161525 (*DC-SIGNR*), rs4251513 (*IRAK4*), rs3024498 (*IL10*), rs2508450 (*IL10RA*), rs16923189 (*PD-L2*), rs17804441 (*PD-L1*), rs1051922 (*IFNB1*), rs12340866 (*JAK2*), rs3771300 (*STAT1*), rs303215 (*IFIT1*), and rs17622656 (*IRF1*).

Our proposed genome polymorphisms/machine learning approach showed an accuracy that was greater than 85%, alongside a sensitivity and specificity of over 80% and 90%, respectively. The results revealed complex multi-loci human genome signatures to severe COVID-19 predisposition using data from 12 key SNPs and 12 innate immune genes, under different genotype compositions, particularly from the IFN pathway (Table 2). These findings support previous observations that suggest the importance of *IFN* and Interferon stimulated genes (*ISGs*) on COVID-19 outcomes. It also shows that the key element for genetically influenced diseases can be the “genome context,” instead of the commonly used single genetic markers, as observed in dengue complex genetics [23] and COVID-19 [8]. The study by Asteris and colleagues (2021) was based on machine learning (ANN) and human genome and biological data for COVID-19 prognosis. The authors, by means of the next-generation sequencing approach, identified, among 381 SNVs and 133 patients, five critical polymorphisms associated with severe COVID-19 in *C3* (2), *THBD* (1), *CFH* (1), and *CFHR1* (1) genes. They used the genome data in combination with gender and patient age data to develop an ANN-predicted mortality architecture (~90%) for severe COVID-19 [8]. However, it is important to note that our prognosis method still has some limitations, especially if we consider that we used hold-out validation which was implemented with a view to later use the SHAP method, and the lack of a methodology to estimate ancestry. Therefore, the population studied was composed of the same geographical area, which contributed to decreasing the influence of ancestry as a bias [42].

### 4.2. Molecular Biology of the Complex COVID-19 Prognosis Classifier

The proposed complex genome polymorphisms COVID-19 classifier is based on key elements of innate immune genes or viral response genes (Table 2, Figure 3). *IFIH1*, *DC-SIGNR*, and *IRAK4* relate to viral recognition (Figure 3, n° 1). A little further, the *L10*, *IL10R*, *PD-L2*, and *PD-L1* genes relate to the IL10 pathway. Meanwhile, *IFNB1*, *JAK2*, and *STAT1* relate to the IFNβ pathway (Figure 3, n° 2), and, finally, *IFIT1* relates to viral blocking. *IRF1* is a transcription factor that stimulates antiviral gene expression (Figure 3, n° 3). All these elements will be discussed below.

#### 4.2.1. Viral Recognition

The interferon-induced helicase-1 (*IFIH1*) gene encodes the melanoma differentiation-associated protein 5 (MDA5), an intracellular pattern recognition receptor (PRR) that is responsible for recognizing viruses [43] and is implicated in autoimmune and autoinflammatory diseases, such as type 1 diabetes (T1D) [57].

*DC-SIGNR* is a DC-SIGN homolog that is encoded by the *CLEC4M* gene with 77% amino acid identity and features polymorphism in its extracellular neck region, which is encoded by tandem repeat domain in exon 4 [44,45]. Together with *ACE2*, *DC-SIGNR* is expressed in the lung and small bowel of patients who are fatally infected with SARS-CoV [58]. A genetic risk association study conducted by Chan et al., 2006 [59] on SARS-infected patients during the outbreak in 2003 showed that individuals homozygous for *CLEC4M* tandem repeats were less susceptible to SARS infection. They also showed that homozygous *DC-SIGNR* cells had a higher capacity to bind SARS-CoV with little dissociation, leading to viral degradation in a proteasome-dependent manner and a lower capacity for trans infection. Incidentally, Interleukin-1 receptor-associated kinase-4 (IRAK-4) is a kinase that activates NF-kappaB in both the Toll-like receptor (TLR) and the T-cell receptor (TCR) signaling pathways [46].

#### 4.2.2. IL10, IFNB and JAK-STAT Pathways

PD-L1 (CD274) and PD-L2 (PDCD1LG2), also known as programmed cell death-1 ligands 1 and 2, are cell-surface receptors that are found on hematopoietic cells. Along with PD-L1, PD-L2 binds to PD-1 (programmed cell death-1 receptor): an inhibitory receptor that acts as an immune checkpoint and plays a role in suppressing the adaptive immune system. Both PD-L1 and PD-L2 inhibit T-cell proliferation and cytokine production to ensure homeostasis and reduce the damage caused by the host immune response [49]. When both PD-L1 and PD-L2 were blocked, dendritic cells enhanced T-cell proliferation and cytokine production, including that of IFN- and IL-10, which showed that they were the negative regulators of the IL-10 pathway [60]. PD-L1 is also a positive regulator of Interferon stimulated genes (ISGs), activating their expression [23].

Interleukin-10 (IL-10) and other anti-inflammatory cytokines have a central role in infection, preventing host damage by limiting immune response, and can be produced because of virus replication. An understanding of how the *IL10* expression is complexly regulated was related by Saraiva and O’Garra, 2010 [47], including upregulation by TLR-independent stimuli, such as DC-SIGN, another target identified in our study. Lu, 2021 [61] related that cytokine storm is similar to COVID-19 and SARS-CoV patients, but in severely ill COVID-19 patients, IL-10 is incredibly elevated, which suggests that IL-10 could be a putative biomarker.

The engagement of the JAK-STAT signaling circuit by the ligation of the IL-10 receptor complex occurs mainly through STAT3 [48,52]. It has been shown that individuals with severe immune-mediated diseases, such as very early onset inflammatory bowel disease (VEOIBD) and autoimmune thyroid diseases, have polymorphisms in *IL-10* and its receptors, *IL-10RA* and *IL-10RB*. *IL-10RA* mutation at the 3′ ends of exon 4 (c.537G → A) reflected an increased risk of severe IAE (influenza-associated encephalopathy) [62]. The *IFNβ* expression, in turn, was the result of the PRRs activation, for example, MDA5 and IRAK4, by a specific PAMP. The signaling begins when it binds with the IFNA receptor and triggers the JAK-STAT pathway, activating interferon-stimulated genes (*ISGs*), such as IFIT1 and IRF1. The upregulation of *ISGs* simulates an antiviral state, which inhibits viral entry, replication, and translation in both non-infected and infected cells, respectively [50,51].

#### 4.2.3. Virus Replication

The interferon-induced protein with tetratricopeptide repeats 1 (IFIT1) inhibits virus replication by binding and regulating the functions of cellular and viral proteins and RNAs [54]. For example, IFIT1 impedes JEV replication by inhibiting mRNA translation through direct binding to mRNA 5′ structures [53]. This is the first report associating rs303215 T>C with a viral disease.

Finally, IRF1 is a transcriptional regulator that was originally characterized in the type I Interferon pathway [55] but is now known as an activator of genes involved in both innate and acquired immune responses [56]. In the early phase of a virus infection, *IRF1* is highly expressed, stimulating the production of *IFNs* and *ISGs*. Then, in the later phase, secreted IFN triggers the STAT1-STAT2-IRF9 pathway to induce IRF1, which acts as a feedback loop [63]. Together and in combination, all these factors must be important to COVID-19′s severe phenotype progress.

## 5. Conclusions

Here, we presented the genome polymorphisms/machine learning approach for predicting severe COVID-19 phenotype based solely on human innate immune genome markers complex data and machine learning techniques: a robust approach for a potential severe COVID-19 prognosis tool.

This method has some key novelties: it can be applied to any genetically influenced disease, in any development stage, even before infection (in case of infectious diseases), using a broad human sample. Moreover, it is a method free of clinical and laboratory data and medical interpretation (that depends on the medical experience). The presented tool was able to select the optimal loci combination data and accurately predict those patients who would develop COVID-19 disease based on their genome background, including key elements of host antiviral response and innate immune system (i.e., avoiding clinical routine limitations).

However, our method presents some limitations that need to be considered: single-center patient group and ancestry considerations, data validation aspects, etc. Despite this, we can consider that our method is a preliminary approach with a potential application in the future clinical routine.

## Figures and Tables

**Figure 1 viruses-15-00645-f001:**
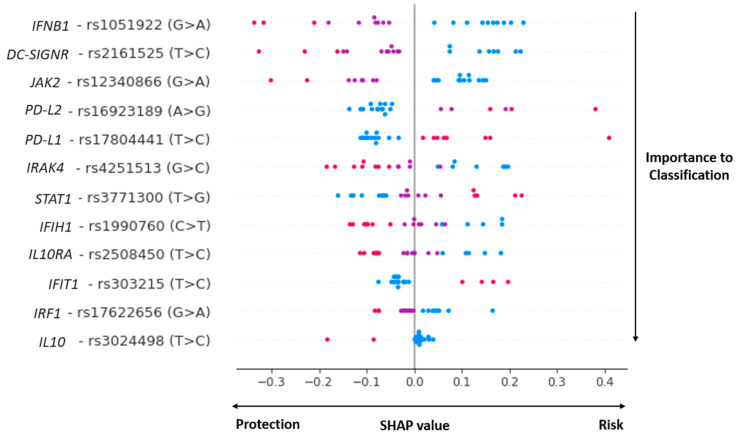
The general impact of each human genotype on genome polymorphisms -ML COVID-19 prognosis classifier under SHAP (Shapley Additive exPlanations) analysis. The impact of each selected genotype (feature) on genome polymorphisms/machine learning model output (mild or severe COVID-19 cases). In blue, reference homozygotes; in purple, heterozygotes; in pink, alternative homozygotes. The figure depicts SHAP analysis over test dataset only–some alleles may not appear with all three values.

**Figure 2 viruses-15-00645-f002:**
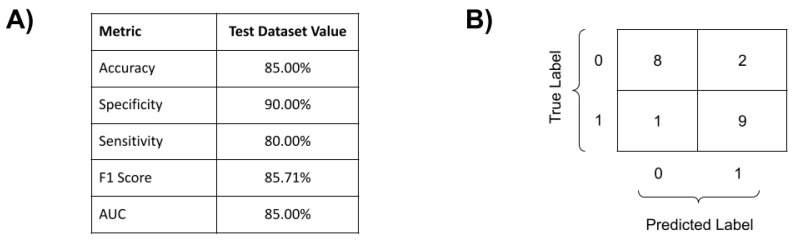
General performance of our human genome polymorphisms/machine learning COVID-19 prognosis classifier (SVM evaluation with Linear Kernel over 20 test samples: 10 mild cases and 10 severe cases). (**A**) Evaluation metrics for our complex COVID-19 classifier. (**B**) Confusion matrix: class 0 stands for mild cases, whereas class 1 stands for severe cases of COVID-19.

**Figure 3 viruses-15-00645-f003:**
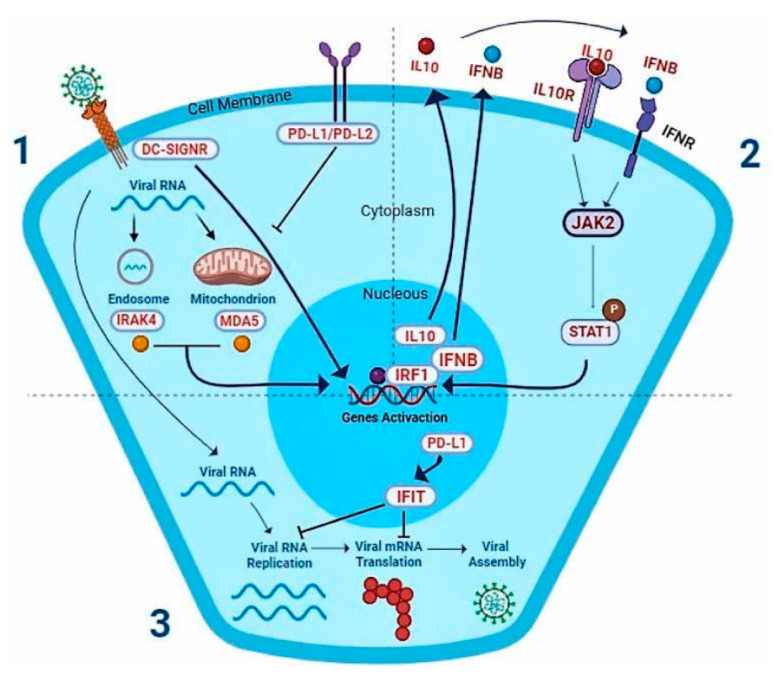
The selected SNPs from the proposed genome polymorphisms/machine severe COVID-19 classifier in cellular context. The genes of the SNPs selected by our classifier are related to IL-10 and IFN cellular pathways, which are highlighted in the figure (in red). In the viral recognition phase (1), the virus is recognized by receptors, such as DC-SIGNR, and then enters the cell, where the vRNA is identified by the intracellular molecules, such as IRAK4 and MDA5 proteins. Then, in the antiviral signaling phase (2), they activate the expression of genes involved in the antiviral response, such as IL-10 and IFN. On the other hand, PD-L1 and PD-L2 are inhibitors of the IL-10 pathway triggered by DC-SIGNR activation. The IL-10 and IFN proteins are delivered to the extracellular compartment, where they connect to their receptors in the cell plasma membrane, activating the JAK-STAT proteins cascade. It triggers the viral blocking phase (3), promoting the expression of antiviral factors that affect viral replication and translation, such as IFIT, as well as positive regulators, including PD-L1 and IRF1. PD-L1, incidentally, activates the expression of other ISGs, maintaining the cell’s antiviral status. While IRF1 is a transcriptional factor that activates the expression of immune genes, such as IFN, PD-L1 and IFIT.

**Table 1 viruses-15-00645-t001:** Brazilian COVID-19 patients group stratification by age and clinical outcomes.

Age (Years)	Male	Female	Mild COVID-19 Patients	Severe COVID-19 Patients
11–21	1 (1.6%)	1 (2.8%)	2 (4.2%)	-
22–30	4 (6.6%)	3 (8.6%)	5 (10.4%)	2 (4.2%)
31–40	11 (18.1%)	4 (11.4%)	3 (6.2%)	12 (25%)
41–50	30 (49.1%)	13 (37.2%)	22 (45.8%)	21 (43.7%)
51–70	15 (24.6%)	12 (34.3%)	14 (29.2%)	13 (27.1%)
71–90	-	2 (5.7%)	2 (4.2%)	-
Total	61	35	48	48

**Table 2 viruses-15-00645-t002:** The best 12 selected SNVs used in our genome polymorphisms/machine learning complex classifier for COVID-19 prognosis.

SNP (RS Code/Alleles)	Function/Location	Gene	Protein Function	References
Viral recognition				
rs1990760 C>T	missense_variant	IFIH1	MDA5 is an intracellular sensor of viral RNA.	[43]
rs2161525 T>C	intron_variant	DC-SIGNR	A C-type lectin that functions in cell adhesion and pathogen recognition.	[44,45]
rs4251513 G>C	intron_variant	IRAK4	A kinase that activates NF-kappaB in both the Toll-like receptor (TLR) and T-cell receptor (TCR) signaling pathways.	[46]
IL-10 and IFN Pathways				
rs3024498 T>C	non_coding_transcript_variant	IL10	A cytokine with pleiotropic effects in immune regulation and inflammation.	[47]
rs2508450 T>C	intron_variant	IL10RA	Receptor for interleukin 10.	[48]
rs16923189 A>G	5_prime_UTR_variant	PD-L2	Negative regulation of interleukin-10 production.	[49]
rs17804441 T>C	intron_variant	PD-L1	Inhibitory receptor-ligand expressed by T-cells and B cells, and various types of tumor cells.	[49]
rs1051922 G>A	“sotp gained”, coding sequence.	IFNB1	A cytokine that belongs to the interferon family of signaling proteins.	[50,51]
rs12340866 G>A	intron variant	JAK2	The non-receptor tyrosine kinase of the JAK/STAT pathway.	[52]
rs3771300 T>G	genic_downstream_transcript	STAT1	Member of the STAT protein family.	[52]
Viral Replication				
rs17804441 T>C	intron_variant	PD-L1	Positive regulator of ISG expression.	[23]
rs303215 T>C	intron_variant	IFIT1	An interferon-induced protein that inhibits viral replication and translational initiation.	[53,54]
rs17622656 G>A	intron_variant	IRF1	A transcriptional regulator that activates the genes involved in both innate and acquired immune responses.	[55,56]

## Data Availability

All data generated in this study are available in the Appendix A section.

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
