# Peer review of "Human Genome Polymorphisms and Computational Intelligence Approach Revealed a Complex Genomic Signature for COVID-19 Severity in Brazilian Patients"

_viruses, 2023, doi:10.3390/v15030645_

Round 1

Reviewer 1 Report

The paper has many flaws, and while I consider it potentially of relevance, these necessitate a major revision of the manuscript.

Major remarks:
- It should be made much more clean what the study adds to what we already know, what is the novelty on which sheds light, given the huge amount of literature already available on the subject (as introduced by the paper itself).
- "SVM-RFE algorithm for feature selection" This should be at least very briefly introduced (but a literature citation is the bare minimum; [26] is not about this method). Also, it would be important to discuss in a few words why this algorithm was chosen for feature selection.
- "The hold-out validation strategy" Why did the authors use hold-out set validation? Cross-validation or bootstrapping would have been better (see 10.1016/s0895-4356(01)00341-9 or 10.1016/j.csda.2009.04.009).
- It seems that the authors *separately* performed a feature selection and the training, validating the later (albeit with an inferior method, see my previous remark, but let's put this aside now). This is not the correct way to perform an internal validation: all steps should be included, i.e., feature selection as well! Both with bootstrap and cross-validation (the two methods that should be used instead of hold-out set validation). Let me quote Frank Harrell in this matter: "Whether using bootstrap or CV, it is imperative that all supervised learning steps be repeated afresh when validating the model. Any analysis that utilized Y including any association-with- Y -based feature selection must be repeated afresh." That means that authors should take a bootstrap sample, perform *both* the feature selection *and* the training on that sample, record performance and then repeat it again and again with new bootstrap samples. (The same goes for cross-validation.) This is necessary, as the present approach of validation is biased: it does not account for feature selection.
- The authors state 6 different machine learning approaches, yet, no data is provided at all about 5 of them!
- Fig 1B should be removed, and the confusion matrix should be presented numerically.
- It is very hard to believe that Fig 1C is indeed the correct ROC curve. Two straight lines connected???

Minor remarks:
- "pandemic with over 53 million cases and over 6 million deaths reported by May 2022 [1]." This should be updated.
- "have described important factors related to the severe progression of COVID-19," 10.3390/v15010175 could be a relevant citation here (more comprehensive than those the authors have presented).
- "191 SNPs from ninety-six Brazilian patients" I don't see why one of the numbers is written out and the other is not.
- "ECMO – extra body ventilation" That should be extracorporeal membrane oxygenation
- "oximetry under 96%," That should be oxygen saturation.
- "P1 (gama),"
- Leading zeros are not needed in Table 1. Also, a percentage should be provided.
- "includes 12 SNPs in 12 innate immune genes (Fig. 1 and Fig. 2)." Figure 1 doesn't say anything about these genes and SNPs.
- "In the conventional analyses, individual OR was analyzed for each variant allele/genotype present in the genome complex classifier (Fig. 1)." I thought authors call univariate association analysis the "conventional analysis" (in contrast to machine learning), but Fig 1. seems to present SVM, a machine learning method. What do they mean by "conventional"?
- Decimal separator should be dot, not comma (this error is repeated several times in the manuscript).

Reviewer 2 Report

The manuscript of Pastor et al. presents a comparison of a traditional statistical approach with an SVM-LK-based approach. The analysis is based on the 96 mildly and severely affected COVID-19-positive participants from two hospitals in Refice, Brasil. The classifier performed relatively well. Also, explainability techniques has been provided. The figures are of suffcient quality.

While it is important to include participants from different ancestries, it is also to remember that this a study including only SNPs (not whole exome or genome sequencing) and a limited number of participants from one city. That is why the model has to be adequately discussed in the context and probably validated on the more extensive data. 

Major comments:

1. Please make the source code available

2. Introduction: Please include information about the architecture of other networks and their properties. There are more AI and deep learning approaches published that are not mentioned here. Maby a table summarizing the main approaches would be useful?

3. Methodology: How were 296 polymorphisms for sequencing chosen? Based on the literature, internal standards?

4. The cohort is quite small. Are there any available data from Brazil or globally to test the network and compare the results?

5. Results: Please include the sex structure of the cohort. Is there any information regarding chronic diseases available?

6. The section Molecular biology of the complex COVID-19 prognosis classifier should be significantly shortened, as this is not the main of the manuscript to characterize it in detail.

7. Please discuss in detail the limitations of the study (single center, one cohort, validation aspect, comparison to approaches with WGS data and not including clinical data, etc.)

8. this is one center data from a small cohort. Can such data be used in clinical practice or do they require further validation? Please state in the Conclusions. I found the current conclusions too forward looking.

Minor comments:

1. Please write the name of genes in italics

2. Would it be possible to describe and maybe present in a graph architecture of the network applied?

3. I find the name "genome/machine learning approach" confusing. It is not based on the whole genome.

4. The part data preprocessing could be shortened, if it is not possible partially added as a supplement.

5. Please discuss shortly the genetic structure of the cohort (ancestry)

6 . SHAP is mentioned for the first time in paragraph 3.2, but there is no reference given there

7. The paragraphs and subparagraphs in the Materials are Methods are numbered, but in the other sections not.

8. Please provide the list of all sequenced SNPs as a supplement.

9. Are the results in line with other similar studies?

10. From the abstract it is not clear that traditional statistical approaches are compared with machine learning

Round 2

Reviewer 1 Report

While I still feel that there are important areas where the research could have been improved, they're now at least mostly ackownledged as limitations, so I can accept the paper with the following minor revisions:

- I ask the authors to add the paragraph on the novelty and relevance of this research (i.e., not to address it only to me).
- Data on the other 5 methods should very definitely be provided, at least as a supplementary material.
- Neither Figure 1A and 1B are actually figures; both are very clearly tables.
- Leading zeros still appear in Table 1 (02, 05, 03, 02).
- "P1 (Gama)," It should be gamma (with two 'm's).

Author Response

Review 1

While I still feel that there are important areas where the research could have been improved, they're now at least mostly ackownledged as limitations, so I can accept the paper with the following minor revisions:

Response: Thank you for your important questions and suggestions. Below we made a list of answers point-by-point.

- I ask the authors to add a paragraph on the novelty and relevance of this research (i.e., not to address it only to me).

Response: Ok, it was taken into account. This information was included in the conclusion section.

- Data on the other 5 methods should very definitely be provided, at least as a supplementary material

Response: We agree: the required data were provided as supplementary material.

- Neither Figure 1A and 1B are actually figures; both are very clearly tables.

Response: Figure 1 was renamed as table 2. And all figures and tables were renamed according to these changes.

- Leading zeros still appear in Table 1 (02, 05, 03, 02).

Response: Sorry for this mistake. It was taken into account and changed in the main document. 

- "P1 (Gama)," It should be gamma (with two 'm's).

Response: The reviewer is entirely correct. The name was corrected in the manuscript.

Reviewer 2 Report

I would like to thank the authors for the revisions. Please integrate them for the next time as the track changes, so that it is easier to check what is changed. 

Major comments: 

1. The major problem is that either the revision was not integrated in the text or I cannot download the most recent version. For example, there is a very well-written limitations paragraph in the cover letter that I cannot localize in the text. Please use the track changes.

2. Name of the network: thank you for changing it. However, I am still not convinced that using the word "dual" makes sense as every network is fed with some data and it does not mean that it is dual. 

3.  Introduction: Thank you for giving an example of the application of AI in the COVID pandemic. Please comment also shortly on the networks architecture.

4. Introduction: several variants are mentioned, but it is not easy to keep an overview of them. A table would help.

5. Methodology-your answer regarding how SNP has been chosen should be integrated into the main text, as this is important. Could you add how many people reviewed the literature, in which year and what database has been used?

5.  Conclusions are too long, but maybe I do not have the proper version. I do not think that the results of the small study based on infectious diseases can be extended to Mendelian diseases. Also, it is not right to recommend using it in the clinical routine, before further validation, larger cohorts, and issue of genetic ancestry. There are many examples when AI approaches were used too early in the clinical routine (e.g. sepsis algorithm). This can bring disadvantages for patients and a health system.

6. There are some genetic studies on the COVID19 in Brasil E.g. https://doi.org/10.1038/s41439-021-00146-w  (Secolin et al, 2021)

The author used exome for their alaysis and cite two publicly accessible databases (however, they require registration). Please add this manuscript to your reference list. I do not know if there is a possibility to access these open databases.

7. Thanks for adding SNPs list. Would it be possible to add the name of the gene as well?

8.  For me, one of the most important aspects is that this study encompasses only Brasil and one region only. On the genetic level, is there anything that distinguishes the Brazilian population from other populations (eg. genetic diversity, diverse ancestry, variants related to infectious diseases)? This does not have to be related to COVID or infectious diseases.

Author Response

Review 2

I would like to thank the authors for the revisions. Please integrate them for the next time as the track changes, so that it is easier to check what is changed.

Response: We appreciate your significant contributions to our work. Below we answer all your questions.

Major comments:

  1. The major problem is that either the revision was not integrated in the text or I cannot download the most recent version. For example, there is a very well-written limitations paragraph in the cover letter that I cannot localize in the text. Please use the track changes.

Response: We are sorry if there was any confusion in our response. As suggested, we kept the changes in the manuscript in the "Track-Changes" mode and added shortly the limitations in the discussion and the conclusions sections.

  1. Name of the network: thank you for changing it. However, I am still not convinced that using the word "dual" makes sense as every network is fed with some data and it does not mean that it is dual.

Response: Ok, It was taken into account. To improve better understanding of our work we changed the title removing the word “DUAL”. The title now is:  HUMAN GENOME POLYMORPHISMS AND COMPUTATIONAL INTELLIGENCE APPROACH REVEALED A COMPLEX GENOMIC SIGNATURE FOR COVID-19 SEVERITY IN BRAZILIAN PATIENTS.

  1. Introduction: Thank you for giving an example of the application of AI in the COVID pandemic. Please comment also shortly on the network's architecture.

Response: As requested, the comments on network`s architecture have been added to the introduction.

  1. Introduction: several variants are mentioned, but it is not easy to keep an overview of them. A table would help.

Response: We thank you for your observation, but we inform you that we have revised the text and believe that it is in logical form. In this sense, we understand that the presentation of a table would not help much in this matter.

  1. Methodology-your answer regarding how SNP has been chosen should be integrated into the main text, as this is important. Could you add how many people reviewed the literature, in which year and what database has been used?

Response: Researchers from the Oswaldo Cruz Foundation and the University of Washington (USA) were directly involved in the discussions to include the targets in our SNPs panel. Queries were made in the literature available at that time, close to the 2010s (see Davi et al 2019, ref. 24). We also inform you that all selected SNPs are available in the NCBI genetic polymorphism database (https://www.ncbi.nlm.nih.gov/snp/).

  1. Conclusions are too long, but maybe I do not have the proper version. I do not think that the results of the small study based on infectious diseases can be extended to Mendelian diseases. Also, it is not right to recommend using it in the clinical routine, before further validation, larger cohorts, and the issue of genetic ancestry. There are many examples when AI approaches were used too early in the clinical routine (e.g. sepsis algorithm). This can bring disadvantages for patients and a health system.

Response:  We agree with you: we edited the conclusions and removed the mentions of Mendelian diseases in the document. We also agree that, at the present time, our work remains preliminary, with a potential for future application after revalidation appointments, considering all the issues raised during this review.

  1. There are some genetic studies on the COVID19 in Brasil E.g. https://doi.org/10.1038/s41439-021-00146-w (Secolin et al, 2021)

The author used exome for their analysis and cite two publicly accessible databases (however, they require registration). Please add this manuscript to your reference list. I do not know if there is a possibility to access these open databases.

Response: Thank you for your suggestion. We cited this study in our manuscript. Regarding the use of their data, we will consider contacting them after our own publication is available. 

  1. Thanks for adding SNPs list. Would it be possible to add the name of the gene as well?

Response:  Yes, of course, we provided the list of genes in the SNPs table (as supplementary material).

  1. For me, one of the most important aspects is that this study encompasses only Brasil and one region only. On the genetic level, is there anything that distinguishes the Brazilian population from other populations (eg. genetic diversity, diverse ancestry, variants related to infectious diseases)? This does not have to be related to COVID or infectious diseases.

Response: We agree that our group of patients brings some limitations to the study. Thus, the limitations related to the studied population were presented in the manuscript and also in the first round of responses, which parts of our answers are reproduced below (maybe there was some confusion in the versions):

"Our present complex genetic approach based on only genomic data and AI, despite being promising, has, at this point, some obvious limitations that must be clarified. The first one is the limited number of patients, all from the city of Recife, Brazil. In this respect, we can consider that our method is still an approach with a “potential application”. At that first moment, we object to testing the ability of machine learning to be able, using only genomic-immunological data, to differentiate patients predisposed to developing severe COVID-19 from non-severe ones, which turned out to be possible, with reasonable statistical margins. In a second moment, as previously mentioned, our intention is to expand the testing of this methodology in a much larger number of patients, from different geographic origins and with different genetic backgrounds, through access to international consortiums of COVID-19 genetics. The initial contacts for this second moment, in fact, have already started. We hope that access to these international networks will allow us to carry out the necessary validations of the proposed methodology."

About our population ancestry: “A limitation of our study was the lack of a methodology to estimate ancestry. Indeed, a study with a panel of 40 validated ancestry-informative insertion–deletion DNA polymorphisms individually estimated the European, African, and Amerindian ancestry components of 934 self-categorized White, Brown, or Black Brazilians from the four most populous regions of the country. This study showed that in all regions investigated, European ancestry was predominant, with proportions ranging from 60.6% in the northeast to 77.7% in the south [43]. Based on these data we could assume that there was most probably a similar proportion of genetic backgrounds in the studied groups” [43].

Thus, we believe that in the population studied, there are no marked genetic differences in terms of exposure to viral diseases. Anyway, we inform you that such questions will be deeply considered in our future work, especially considering the number of patients, ancestry, etc.